# β_2_E153 Residue at Loop B of GABA_A_R Is Involved in Agonist Stabilization and Gating Properties

**DOI:** 10.3390/ijms27010047

**Published:** 2025-12-20

**Authors:** Michał A. Michałowski, Aleksandra Brzóstowicz, Jerzy W. Mozrzymas

**Affiliations:** Department of Biophysics and Neuroscience, Wroclaw Medical University, 50-368 Wrocław, Poland

**Keywords:** GABA, ligand, gated, receptor, patch clamp, model, kinetics, mutation, structure

## Abstract

γ-Aminobutyric acid type A receptors (GABA_A_Rs) are pentameric ligand-gated ion channels mediating fast inhibitory neurotransmission in the mammalian brain. Although recent structural and kinetic studies have advanced understandings regarding their activation mechanisms, the molecular determinants coupling agonist binding to channel gating remain unclear. We investigated the contribution of the β_2_E153 residue, located on loop B of the extracellular domain, to the activation of α_1_β_2_γ_2_ GABA_A_Rs. Macroscopic and single-channel patch clamp recordings were used to characterize two β_2_E153-mutants: charge reversal (β_2_E153K) and hydrophobic substitution (β_2_E153A). Both substitutions disrupted normal receptor kinetics, with β_2_E153K selectively accelerating deactivation and β_2_E153A affecting both deactivation and desensitization. Single-channel analysis showed that β_2_E153A reduced open probability and mean open times, consistent with altered gating transitions inferred from kinetic modeling. Structural inspection suggested that β_2_E153 forms electrostatic interactions with β_2_K196 and β_2_R207 to stabilize loop C and maintain the agonist-bound conformation. The disruption of this interaction likely destabilizes loop C, leading to weakened agonist binding and modified gating. Overall, our results identify β_2_E153 as a key element in the long-range allosteric network linking the binding site to the channel gate in GABA_A_Rs.

## 1. Introduction

γ-Aminobutyric acid type A receptors (GABA_A_Rs) are pentameric ligand-gated ion channels (pLGIC) that mediate synaptic inhibition in the mammalian brain. Their activation by GABA produces chloride influx and, under physiological conditions in adults, hyperpolarization of the postsynaptic membrane (for review see [1]). Dysfunctions in GABA_A_Rs are linked to a wide spectrum of neurological and psychiatric disorders, including epilepsy, anxiety, autism, and schizophrenia, and these receptors are major pharmacological targets for benzodiazepines, barbiturates, anesthetics, and neurosteroids [2,3,4,5,6].

The canonical synaptic GABA_A_R is assembled from two α, two β, and one γ or δ subunit, with the α_1_β_2_γ_2_ combination being the most prevalent in the adult brain [7] (Figure 1a). The orthosteric binding sites for GABA are located at the β^+^/α^−^ (primary/complementary) subunit interfaces in the extracellular domain (ECD, Figure 1a,b, for review on GABA_A_R structure see [1,8,9,10]). Upon agonist binding, the receptor undergoes a sequence of conformational changes, including preactivation (“flipping”), channel opening, and transitions into desensitized states. These processes are governed by complex allosteric communication between the ECD and transmembrane domain (TMD, Figure 1b). Despite substantial progress in the structural biology and electrophysiology of this receptor, the precise molecular determinants coupling agonist binding to gating remain incompletely understood.

The β_2_E153 residue is located on the loop B between loop A and β-strands 9 and 10 forming loop C. Residues β_2_E155 and α_1_R66 located above β_2_E153 form strong and direct electrostatic interaction with the GABA molecule (visible in structural data, e.g., Kim et al., 2020 [11]), indicating the possible involvement of the β_2_E153 in the receptor activation by interaction with the β_2_E155 (Figure 1b). Some mutations of the latter residue strongly impair GABA binding, increasing EC_50_ value by several orders of magnitude [12], and others fail to saturate even at 300 mM GABA [13,14], underlining the role in agonist binding. It has been additionally shown that mutation of this amino acid induces marked spontaneous activity, indicating its involvement in receptor gating [12]. Jatczak-Śliwa et al. [13] addressed its effect on gating by recording macroscopic and single-channel currents, and provided evidence that the β_2_E155 mutation indeed strongly affected the receptor gating, primarily in terms of preactivation and, to a smaller extent, also desensitization. On the other hand, Venkatachalan et al. [15] showed that β_2_E153 forms a salt bridge with β_2_K196, suggesting a different interaction partner of β_2_E153 in its role in the GABA_A_R activation process.

Considering that β_2_E153 and β_2_E155 are located distantly from the channel gate (approximately 50 Å), it would be interesting to check if the former residue is involved in the receptor gating, similar to β_2_E155. In our recent study [16], we proposed that “gating is global and binding is local”, and the impact of β_2_E155 mutations on gating confirms the former part of this rule. Considering these premises, it seems interesting to provide a detailed analysis of how mutation of β_2_E153 affects the binding and gating of the GABA_A_ receptor. In our study we focused on β_2_E153K (charge reversal) and β_2_E153A (hydrophobic) mutants to provide an accurate assessment regarding to what extent binding and gating processes are affected by the β_2_E153 mutation.

Here, we show that mutations at β_2_E153 markedly alter the receptor’s function, with β_2_E153K selectively accelerating deactivation, and β_2_E153A affecting both deactivation and desensitization kinetics in macroscopic measurements (Figure 2). Single-channel analysis further demonstrates that β_2_E153A reduces open probability and shortens mean open times, and these effects are accompanied by shifts in shut time distributions (Figure 3). Kinetic modeling indicates that these phenotypes can be parsimoniously explained by changes in binding and gating (particularly opening, β, and desensitization, d) rate constants, with accelerated agonist unbinding and altered gating transitions emerging as key mechanisms (Figure 4 and Figure 5). Together, our findings underscore the pivotal role of loop B, and specifically β_2_E153, in the stabilization of the agonist in the binding site and receptor gating signal transduction.

**Table 1 ijms-27-00047-t001:** Parameters of the macroscopic currents mediated by WT and β_2_E153-mutant GABA_A_Rs. Control values (WT) in italics; statistically significant different values (*p*-value < 0.05) in bold.

Receptor	EC50	RT	τ_des fast_	τ_des slow_	A%_des fast_	A%_des slow_	FR10	FR300	FR500	τ_dea_
	[μM]	[ms]	[ms]	[ms]						[ms]
WT	*47.07*	*0.51*	*3.22*	*130.69*	*0.58*	*0.18*	*0.28*	*0.22*	*0.2*	*407.68*
		*±0.07*	*±0.27*	*±14.44*	*±0.01*	*±0.01*	*±0.03*	*±0.02*	*±0.02*	*±11.07*
		*n = 5*	*n = 5*	*n = 5*	*n = 4*	*n = 5*	*n = 5*	*n = 5*	*n = 5*	*n = 5*
β_2_E153K	232.37	0.56	2.62	130.81	0.65	0.16	0.29	0.2	0.16	**74.37**
		±0.04	±0.15	±15.06	±0.01	±0.01	±0.02	±0.01	±0.01	**±6.67**
		n = 6	n = 7	n = 7	n = 7	n = 7	n = 7	n = 7	n = 7	**n = 6**
β_2_E153A	206.73	0.53	2.68	194.61	**0.79**	0.15	**0.19**	**0.12**	**0.1**	**72.28**
		±0.03	±0.25	±32.58	**±0.01**	±0.01	**±0.02**	**±0.01**	**±0.01**	**±8.35**
		n = 7	n = 8	n = 8	**n = 6**	n = 8	**n = 8**	**n = 7**	**n = 8**	**n = 8**

**Table 2 ijms-27-00047-t002:** Parameters of the single-channel currents mediated by WT and β_2_E153-mutant GABA_A_Rs. Control values (WT) in italics; statistically significant different values (*p*-value < 0.05) in bold.

Receptor	p-Open	Shut Times									Open Time
		P_1_	P_2_	P_3_	P_4_	τ_1_	τ_2_	τ_3_	τ_4_	τ_mean_	τ
						[ms]	[ms]	[ms]	[ms]	[ms]	[ms]
WT	0.77	*0.67*	*0.27*	*0.05*	*0.005*	*0.03*	*0.22*	*1.35*	*27.14*	*0.3*	*1.75*
	±0.02	*±0.03*	*±0.04*	*±0.01*	*±0.001*	*±0.01*	*±0.02*	*±0.13*	*±6.33*	*±0.06*	*±0.04*
	n = 5	*n = 5*	*n = 5*	*n = 5*	*n = 5*	*n = 5*	*n = 4*	*n = 5*	*n = 5*	*n = 5*	*n = 3*
β_2_E153K	0.75	**0.57**	0.32	0.08	0.003	0.04	0.29	1.48	14.33	0.27	1.53
	±0.01	**±0.03**	±0.01	±0.02	±0.001	±0.01	±0.02	±0.34	±1.28	±0.02	±0.10
	n = 5	**n = 5**	n = 4	n = 5	n = 4	n = 5	n = 5	n = 5	n = 4	n = 5	n = 5
β_2_E153A	**0.5**	**0.52**	0.32	0.14	0.027	**0.05**	0.26	1.28	20.62	**0.79**	**1.12**
	**±0.02**	**±0.02**	±0.02	±0.02	±0.001	**±0.01**	±0.01	±0.06	±1.47	**±0.08**	**±0.08**
	**n = 5**	**n = 5**	n = 5	n = 5	n = 3	**n = 5**	n = 4	n = 4	n = 5	**n = 5**	**n = 5**

**Table 3 ijms-27-00047-t003:** Parameters of the macroscopic kinetic model of the WT GABA_A_R.

k_on_ [1/mM ms]	10.00	α [1/ms]	1.20
k_off_ [1/ms]	0.90	d_2_ [1/ms]	20.00
δ [1/ms]	5.00	r_2_ [1/ms]	0.15
γ [1/ms]	4.00	d_2_’ [1/ms]	1.00
β [1/ms]	25.00	r_2_’ [1/ms]	0.005

**Table 4 ijms-27-00047-t004:** Parameters of the steady state kinetic model of the WT and β_2_E153-mutant GABA_A_Rs. Control values (WT) in italics; statistically significant different values (*p*-value < 0.05) in bold.

Receptor	δ [1/ms]	γ [1/ms]	β_2_ [1/ms]	α_2_ [1/ms]	d_2_[1/ms]	r_2_ [1/ms]	d_2_’ [1/ms]	r_2_’ [1/ms]
WT	*6.14*	*7.76*	*27.25*	*1.49*	*1.17*	*0.87*	*0.12*	*0.09*
	*±0.31*	*±1.43*	*±0.85*	*±0.12*	*±0.23*	*±0.10*	*±0.03*	*±0.01*
	*n = 4*	*n = 5*	*n = 5*	*n = 5*	*n = 5*	*n = 5*	*n = 5*	*n = 5*
β_2_E153K	5.34	5.17	**17.27**	1.41	1.09	0.99	0.07	0.1
	±0.48	±0.03	**±1.72**	±0.11	±0.36	±0.23	±0.02	±0.02
	n = 5	n = 3	**n = 5**	n = 5	n = 5	n = 5	n = 5	n = 5
β_2_E153A	5.6	6.16	**14.14**	1.51	2.05	0.86	**0.3**	0.06
	±0.17	±0.05	**±1.79**	±0.11	±0.14	±0.09	**±0.03**	±0.01
	n = 4	n = 3	**n = 5**	n = 5	n = 4	n = 5	**n = 5**	n = 5

## 2. Results

### 2.1. β_2_E153 Mutations Affect Deactivation and Macroscopic Desensitization of the GABA_A_R

Our investigation into the role of the β_2_E153 residue located in loop B of the GABA_A_R began with an analysis of concentration–response relationships for wild-type (WT) α_1_β_2_γ_2_ receptors and selected mutants, using whole-cell electrophysiological recordings. Both the β_2_E153A and β_2_E153K mutations induced a rightward shift in the dose–response relationship, increasing the EC_50_ by approximately five-fold, without affecting the Hill slope coefficient (Figure 2a). In all cases, 10 mM of GABA was sufficient to achieve full receptor activation and saturate the current response. Thus, these data show that the impact of the β_2_E153 mutation on the binding properties of the receptor is relatively minor, and is orders of magnitude smaller compared to that observed for the β_2_E155 mutation; although, as we show below, its impact on current responses is still clearly observable.

To achieve a more detailed kinetic picture of the functional consequences of β_2_E153 mutations, macroscopic current recordings were performed in excised outside-out patches using an ultrafast solution exchange system at saturating GABA concentrations. The representative traces are shown in Figure 2b. The recorded currents were analyzed to extract the kinetic parameters characterizing the current time course: the rise time (Figure 2c); the macroscopic desensitization time constants for fast and slow components (Figure 2d,e) together with their percentages (Figure 2f); the remaining current fractions at 10, 300, and 500 ms after agonist application (Figure 2g); and the deactivation time constant (Figure 2h). The numerical values of the data are presented in Table 1.

The most prominent effect of the β_2_E153 mutations was observed in the deactivation phase, which was accelerated approximately five-fold in both β_2_E153A and β_2_E153K compared to WT (Figure 2h). Additionally, β_2_E153A, but not β_2_E153K, altered the desensitization profile. In β_2_E153A, the percentage of the fast desensitization component increased (Figure 2f) and reduced the remaining current fraction throughout the examined agonist application times (Figure 2g). In contrast, neither mutation affected the rise time of the current following agonist application (Figure 2c).

In summary, the β_2_E153K mutation selectively accelerated deactivation kinetics, whereas β_2_E153A affected both deactivation and macroscopic desensitization, indicating a stronger alteration of receptor gating.

### 2.2. Influence of the β_2_E153 Mutations on the Receptor’s Open and Shut Dwell Times

To further investigate the effects of β_2_E153 residue mutations, single-channel recordings were performed in the steady-state conditions using saturating [GABA] concentration (10 mM). This approach enables a more detailed kinetic analysis of the impact of the mutations on receptor gating, being considerably less sensitive to overparameterization than the analysis of the macroscopic currents [1,17]. The representative single-channel traces are shown in Figure 3a. The results of the single-channel analysis including the open and shut time distributions and the open and shut mean times are presented in Figure 3a–f and the numerical values of presented data are presented in Table 2.

The mean open probability within clusters of channel activity was significantly reduced in the β_2_E153A mutant, whereas it remained comparable to WT values in the β_2_E153K mutant (Figure 3b). In clusters, at the saturating GABA concentration, receptors are assumed to be in the fully bound state, and thus the changes in the open probability indicate alterations in the receptor transitions other than binding and do not directly reflect the alterations in the EC_50_ value. In both mutants, the amplitudes of the single-channel currents were comparable to those of the WT receptor, indicating that the mutations did not alter the channel conductance. In addition, in the case of mutations, no conductance sublevels or extra-large current amplitudes were observed. The distribution of the shut times was best described by four exponential components, while the open times were fitted well with a single component. The representative dwell time histograms are presented in Figure 3c (open times) and Figure 3e (shut times). The mean open time (corresponding to the single component) was significantly shortened in the β_2_E153A mutant, with no change observed in β_2_E153K (Figure 3d). Similarly, the mean shut time (calculated across all four weighted components) was significantly prolonged in β_2_E153A, but remained unaffected in β_2_E153K (Figure 3f).

A more detailed analysis of the shut-state dwell times is shown in Figure 3g,h. In the case of the β_2_E153A mutant, the mean duration of the shortest shut component (τ_shut1_) was prolonged and its percentage was reduced (Figure 3g,h). Concurrently, the percentage of the third shut-time component was increased, contributing to the overall prolongation of the mean shut time (Figure 3f). Additionally, the total contribution of the shortest shut-time component in the distribution was decreased in the β_2_E153K mutant similarly to β_2_E153A (Figure 3h).

Macroscopic recordings under dynamic conditions revealed a pronounced effect of both β_2_E153 mutations on deactivation kinetics. In the case of the β_2_E153A mutant, these changes were accompanied by alterations in macroscopic desensitization, yet with this type of recording it is difficult to precisely indicate which specific receptor’s features were affected, giving rise to this phenotype. Considering single-channel analyses, these effects may be attributed to a prolonged duration of the shortest shut state, a shortened open-state dwell time, and shifts in the relative contributions of the individual shut-state components. In contrast, the β_2_E153K mutation affected only the deactivation phase of macroscopic currents, with no significant impact on the macroscopic desensitization kinetics. This observation was consistent with single-channel data, where only minor changes were observed in the relative contributions of shut-state components, and no significant changes in open or shut durations were detected, indicating a relatively minor impact on the receptor gating.

### 2.3. Impact of β_2_E153 Mutations on the Ligand Binding and Receptor Gating

To further elucidate the role of loop B and the β_2_E153 residue in the gating mechanism of the GABA_A_R, kinetic modeling was performed. A Markov model, incorporating sequential agonist binding steps (at both GABA binding sites), a doubly bound flipped state, an open state, and two desensitized states (Figure 4a), was used to simulate the receptor responses to 500 ms GABA pulses at a saturating concentration of 10 mM. The initial model was optimized to reproduce the kinetics of the WT receptor (rate values in Table 3). Subsequently, individual rate constants were systematically varied (from 5% to 1000% of their baseline values) to probe different scenarios (Figure 4b,c), and corresponding macroscopic current traces were generated (Figure 4c). For each simulation, kinetic parameters analogous to those obtained from experimental recordings were extracted, and their dependence on the modified rate constant was plotted (Figure 4d).

The modeling revealed that the most prominent effect of β_2_E153 mutations—accelerated deactivation—could be replicated by a number of scenarios postulating changes in several rate constants (Figure 4b). The potential mechanisms included: (i) decreased desensitization rates (d, d′) and/or increased resensitization rates (r, r′), (ii) a decreased flipping rate (δ) and/or an increased unflipping rate (γ), or (iii) an increased unbinding rate (k_off_). However, changes to gating rates had only a minor impact on deactivation kinetics.

Among these scenarios, a reduction in desensitization rates is unlikely, as macroscopic desensitization was unaltered in the β_2_E153K mutant and enhanced in β_2_E153A. Similarly, modifications to flipping/unflipping rates are improbable, as alterations in the flipped state equilibrium would be expected to affect the rising phase of the current response, which remained unchanged for both mutants. Therefore, an increase in the unbinding rate (k_off_) appears to be the most plausible explanation (Figure 4b–d). According to the simulations, the approximate 5-fold decrease in the deactivation time constants can be modeled by the roughly 10-fold increase in the k_off_ rate value (Figure 4c,d). In addition, this rate alteration also increases the EC_50_ value by 5-fold, in agreement with value estimated in the macroscopic measurements (Figure 2a), indicating the specific effect of the mutation on the unbinding process and agonist stability in the binding site.

These changes in binding kinetics can fully account for the phenotype of the β_2_E153K mutant. However, in the case of β_2_E153A, the enhancement of fast desensitization observed experimentally suggests additional effects related to the receptor gating. The most parsimonious explanation is a reduction in the gating opening rate β, which would not only contribute to the increased macroscopic desensitization but would also support the observed acceleration in deactivation kinetics. The major problem with kinetic investigations based on macroscopic recordings is that each feature of current response (e.g., onset, macroscopic desensitization, or deactivation) may depend potentially on all the rate constants considered in the model. This hypothesis, thus, requires further evidence; for this purpose we performed model simulations of the single-channel data, which yielded estimations of the entire set of the gating rate constants.

Since all single-channel recordings were obtained at equilibrium under saturating [GABA], a simplified model was applied in which the agonist-binding steps were omitted (Figure 5a). All idealized single-channel recordings (WT and mutants; *.scn files in DCPROGS package, https://github.com/DCPROGS, accessed on 2 February 2024) were considered to perform the modeling of our data. Using HJCFIT software (https://github.com/DCPROGS/HJCFIT, accessed on 2 February 2024), we have optimized all the gating parameters of the model (Figure 5a) for each analyzed single-channel recording (selected cluster). After having completed the optimization of the rate constants, the dwell time distributions were generated for the respective models (Figure 5b), which matched the experimentally determined dwell time histograms well (Figure 3). The rate constants optimized using HJCFIT software are presented in Figure 5c and Table 4. For both mutants, β_2_E153A and β_2_E153K, a significant decrease in the opening rate β was observed, although for the latter mutation this change showed a tendency to be smaller. This effect is in qualitative agreement with the hypothesis proposed based on macroscopic-level modeling. Additionally, for β_2_E153A, an increase in the slow desensitization rate (d’) was also detected in the single-channel modeling, confirming that for this mutant, alterations in the receptor gating were more pronounced than in the case of β_2_E153A.

In summary, Markov model–based investigations integrating both macroscopic and single-channel data demonstrated that residue β_2_E153 plays a significant role in both binding and gating processes, although they pointed to a stronger gating effect for the β_2_E153A mutant.

## 3. Discussion

In the present study we demonstrate that the β_2_E153 residue is involved in both the agonist binding and gating of the GABA_A_ receptor. The dose–response relationships revealed that GABA potency is reduced in these mutants as the EC_50_ value is increased by approximately 5-fold (Figure 2b). Interestingly, this effect on the agonist binding is by far weaker than that observed for the mutation of the β_2_E155 residue, which neighbors the β_2_E153 and is known to interact with it [15]. Indeed, the EC_50_ determined for mutations of β_2_E153 and β_2_E155, which differ by roughly four orders of magnitude. This observation confirms the above-mentioned rule that the molecular mechanisms determining the agonist binding relies primarily on local interactions. On the other hand, the impact of the β_2_E153 mutation on the agonist binding cannot be regarded as negligible. Considering that a marked shortening of the deactivation kinetics (roughly five times, Figure 2h) related to the β_2_E153 mutation is mostly due to increased unbinding rate k_off_ (impact of altering the opening rate β on deactivation mutating is minimal, Figure 4b–d), it is expected that such a change in receptor functioning would markedly affect the time course of GABAergic currents. Indeed, it is known that the duration of inhibitory synaptic current duration is reflected mostly by the kinetics of the deactivation process [18].

Interestingly, in the case of the β_2_E153 mutations, this marked acceleration of the deactivation was not coupled with decreased desensitization (Figure 2b–h) as was observed in the case of the mutations of many residues located at different regions of the receptor, for example, in the ECD/TMD interface [19] or at the bottom of TMD [20]. The explanation of this coupling is straightforward: desensitization, which is a long-lived shut state, may cause the receptor to be trapped in this state even after the end of agonist application, resulting in a prolongation of the deactivation process. Weak desensitization is therefore typically associated with fast deactivation kinetics [18]. Thus, the altered time course of the deactivation is often just a side product of the changes in the desensitization kinetics However, in the β_2_E153A mutation, the desensitization was enhanced (and would result in slower deactivation) and in β_2_E153K, it was not affected, indicating a specific role of this residue in the stabilization of the agonist in the binding site.

A more precise single-channel analysis indicated, however, that also at equilibrium at saturating [GABA], the kinetics of the receptor activity is affected in case of the β_2_ E153K—the receptor’s open time was decreased and its shut time was increased (Figure 3). These results allowed for the estimation of the parameters of the kinetic model describing GABA_A_R activity (Figure 5a). A decrease in the value of the opening rate β for this mutation pointed to the clear alteration of the gating (Figure 5c). Interestingly, the alanine mutation caused a considerably stronger effect on the receptor gating as the decrease in the opening rate β was larger for β_2_E153A and, additionally, we observed a significant increase in the d’ slow desensitization rate. The reason for which alanine mutation produces a markedly stronger effect on gating than lysine is not clear. As already mentioned, the effect of the β_2_E153 mutation on gating was considerably weaker than that of β_2_E155 [13], indicating that local interactions with agonist are also important in determining the gating properties. On the other hand, it needs to be emphasized that the significant alteration of the receptor gating by the β_2_E153 mutation further confirms that GABA_A_ receptor gating relies on a global network of interactions, as this residue is located very distantly from the channel gate (approx. 50 Å).

According to structural data [11], β_2_E153 is separated from direct contact with both the agonist molecule and β_2_E155 by two charged residues: β_2_K196 and β_2_R207 (β9 and β10 strands). On the sides, β_2_E153 is neighbored by β_2_S209 and β_2_L99 and on the bottom by β_2_T151 and β_2_N100. The interaction with β_2_K196 by salt bridge formation and also with β_2_R207 and β_2_E153 was confirmed by [15]. The specific role of the β_2_R207 in ligand binding/unbinding processes were also showed in a study by Goldschen-ohm et al. (2011) [21]. Each of these residues is located at different β-strand (7, 9 or 10) of the β_2_ subunit ECD (Figure 1b and Figure 6). Thus, we may expect that β_2_E153, located at the most inner β-strand of principal subunit (β7, loop B), locks the loop C (connecting strands β9 and β10, highly mobile; [22,23]) in position, capping the binding site by interaction with β_2_K196 at the most outer of the β-strands (β9) or β_2_R207 from the β-strand located in the middle (β10). Mutation to lysine or alanine would prohibit these interactions, inducing lower stability of the loop C. To verify this hypothesis we prepared the model of the β_2_E153A mutant using the WT structure [11], and estimated the changes in the interaction energy between fragments of the β-strands with residue β_2_E153 and β_2_K196 using FoldX [24] suite. As expected, in the case of each β_2_ subunit, after the mutation to alanine, this interaction was roughly two times less energetically favorable. This would affect the agonist molecule interaction with β_2_E155 and other residues important for binding, like, e.g., β_2_ F200, located at loop C [23], that would explain the observed acceleration of the receptor deactivation (Figure 2h). The explanation of the molecular mechanisms of the effect of β_2_E153A on gating is not straightforward, and may be connected to the interaction with other residues in addition to β_2_E153, β_2_K196, and β_2_R207. For example, mutations of another neighbor of β_2_E153 (Figure 6a), β_2_L99, induced spontaneous activity, indicating its role in the gating process [25].

Another interesting question is whether the proposed above mechanism of the interaction between residues located at respective β strands below ligand-binding β_2_E155 is conserved at other GABA_A_R subunits or is specific for ligand-binding β_2_. The α subunit in the α_1_β_2_γ_2_ GABA_A_R assembly (Figure 1a) forms two interfaces: α_1_/β_2_ (not binding any ligands) and α_1_/γ_2_ (binding modulators). In the case of this subunit (Figure 6b), glutamate at position 155is replaced with glycine (charge removal), glutamate at position 153 with lysine (charge switch), lysine at position 196 with glycine (α_1_, charge removal), or glutamate (α_2–6_, charge switch) and arginine at 207 with valine (also removing charge). Thus, most of the electrostatic interactions are removed (including the key ligand-binding β_2_E155 homolog) and only the homologous interaction between strands β7 and β9 in the case of α_2–6_ subunits types is preserved, but with switched charges. In the γ_2_ subunit (not forming a binding interface as principal subunit) the homologs of β_2_E155 and β_2_E153 are preserved, but homologs of other residues have either switched charges or no charge, so the proposed net of interactions is not preserved (Figure 6b). Interestingly, ρ-type subunits that form functional homomers (each subunit contributing to the binding site) have those residues mostly preserved. In addition, many residues neighboring the analyzed ones are highly conserved among the GABA_A_R subtypes (Figure 6b), underlining the specific role of not conserved β_2_E153, β_2_E155, β_2_K196, and β_2_R207 in ligand binding at the β_2_ subunit.

In conclusion, our findings identify β_2_E153 as an important player in the binding and gating properties of GABA_A_R, and propose that this residue couple agonist binding to gating in the GABA_A_R through electrostatic interactions that stabilize loop C and the bound agonist. Mutations at this site accelerate deactivation and alter gating kinetics, demonstrating that even residues distant from the channel pore can critically influence receptor activation dynamics. These insights into long-range allosteric coupling within the binding domain may guide the rational design of drugs that fine-tune inhibitory signaling by selectively targeting the gating or deactivation mechanisms of GABA_A_Rs.

## 4. Materials and Methods

### 4.1. Cell Culture and Transfection

Human embryonic kidney 293 (HEK293) cells (European Collection of Authenticated Cell Cultures, Salisbury, UK) were used for all experiments. Cells were cultured in Dulbecco’s Modified Eagle’s Medium (DMEM), supplemented with 10% fetal bovine serum (FBS) and 1% penicillin/streptomycin (Thermo Fisher Scientific, Waltham, MA, USA) and maintained in a humidified incubator at 37 °C with 5% CO_2_. Prior to transfection, cells were detached from culture flasks and replated onto poly-D-lysine-coated coverslips (1 μg/mL; Sigma-Aldrich (St. Louis, MO, USA)) placed in 35 mm dishes (Carl Roth, Karlsruhe, Germany).

Transient transfection was performed 24–48 h before electrophysiological recordings using FuGENE HD transfection reagent (Promega, Madison, WI, USA) at a 1:3 DNA:reagent ratio (μg:μL). Plasmids encoding rat GABA_A_R subunits α_1_, β_2_, and γ_2L_ were cloned into a cytomegalovirus (CMV) promoter-based vector (pCMV). For wild-type (WT) receptor expression, cDNA was mixed at a subunit ratio of 1:1:3 (0.5:0.5:1.5 μg for α_1_:β_2_:γ2_L_, respectively). To compensate for the reduced current amplitudes observed in β_2_E153-mutant receptors, the ratio was adjusted to 1:3:3 (α_1_:β_2_E153:γ_2_L). Additionally, 0.5 μg of a plasmid-encoding enhanced green fluorescent protein (EGFP) was included to facilitate the identification of transfected cells.

Transfected cells were visualized using a 470 nm fluorescence illuminator (CoolLED, Andover, UK) mounted on a Leica DMi8 inverted microscope (Leica Microsystems, Wetzlar, Germany).

### 4.2. Macroscopic Electrophysiological Measurements and Data Analysis

Macroscopic current recordings were obtained using the patch clamp technique in the outside-out excised patch configuration (for kinetic analysis) or whole-cell configuration (dose–response analysis). Recordings were performed at room temperature (20–22 °C), 24–48 h post-transfection. Membrane currents were low-pass filtered at 10 kHz and recorded at a holding potential of −40 mV using an Axopatch 200B amplifier (Molecular Devices, Sunnyvale, CA, USA). Signal acquisition was carried out with a Digidata 1550A digitizer and pClamp software, version 10.7 (Molecular Devices, San Jose, CA, USA).

Patch pipettes were pulled from borosilicate glass capillaries (outer diameter: 1.5 mm; inner diameter: 1.0 mm; Hilgenberg, Malsfeld, Germany) using a P-97 horizontal puller (Sutter Instrument, Novato, CA, USA). Pipettes were filled with intracellular solution containing the following (in mM): 137 KCl, 1 CaCl_2_, 2 ATP-Mg, 2 MgCl_2_, 10 K-gluconate, 11 ethyleneglycol-bis(β-aminoethyl)-N,N,N′,N′-tetraacetic acid (EGTA), and 10 2-[4-(2-hydroxyethyl)piperazin-1-yl]ethane-1-sulfonic acid (HEPES); the pH was adjusted to 7.2 with KOH. Pipette resistances ranged from 3 to 5 MΩ. The extracellular (bath) solution contained the following (in mM): 137 NaCl, 5 KCl, 2 CaCl_2_, 1 MgCl_2_, 20 glucose, and 10 HEPES; pH was adjusted to 7.2 with NaOH.

The rapid application of agonist-containing solutions was achieved using a theta-glass [26] pipette mounted on a piezoelectric-driven translator (Physik Instrumente, Karlsruhe, Germany). The open-tip solution exchange times ranged from 150 to 250 μs, depending on the size of the theta-glass tubing and flow rate. A high-precision SP220IZ syringe pump (World Precision Instruments, Sarasota, FL, USA) was used to simultaneously deliver two solutions—one containing GABA and the other a control wash solution—through the separate channels of the theta-glass capillary. All chemicals were purchased from Sigma-Aldrich (St. Louis, MO, USA), unless otherwise specified.

Kinetic analysis of the recorded traces was performed using ClampFit software, version 10.7 (Molecular Devices, San Jose, CA, USA). The onset phase was analyzed using a built-in function calculating the time of the amplitude rise from a 10% up to 90% giving parameter called rise time (RT). The macroscopic desensitization phase was fitted with biexponential function $f_{des}\left(t\right)=A_{des\;fast}e^{-\;\frac{t}{\tau_{des\;fast}}}+A_{des\;slow}e^{-\;\frac{t}{\tau_{des\;slow}}}+C_{des}$, giving time constants (τ_des fast_ and τ_des slow_), and amplitudes (A_des fast_ and A_des slow_) of both components. These amplitudes were normalized to give relative fractions of both components (A%_des fast_ and A%_des slow_): ${A\%}_{des\;fast}=\frac{A_{des\;fast}}{A_{des\;fast}+A_{des\;slow}+C_{des}}$ and ${A\%}_{des\;slow}=\frac{A_{des\;slow}}{A_{des\;fast}+A_{des\;slow}+C_{des}}$. In addition, macroscopic desensitization was also described with a parameter called fraction remaining (FR), defined as the amplitude value of the current after given time from the peak value relative to this maximal value ${FR}_{xxx}=\frac{{Amp}_{xxx}}{{Amp}_{max}}$. Lastly, the deactivation phase was fitted with the singe exponential function, producing the parameter called deactivation time constant (τ_dea_,): $f_{dea}\left(t\right)=A_{dea}e^{-\;\frac{t}{\tau_{dea}}}$. To estimate the EC_50_ values of the amplitudes (A_max_ and A_min_, normalized) of the recorded currents at various GABA concentrations, they were fitted with the four parameter log-logistic equation $A=A_{min}+\frac{A_{max}+A_{min}}{1+({\frac{{EC}_{50}}{\left[GABA\right]})}^{n_{H}}}$ using the Python script (https://github.com/michal2am/bioscripts, accessed on 2 February 2024).

### 4.3. Single-Channel Electrophysiological Measurements and Data Analysis

Single-channel recordings were obtained in the cell-attached configuration at a holding potential of 100 mV. Currents were amplified with an Axopatch 200B amplifier (Molecular Devices, Sunnyvale, CA, USA) at a holding pipette potential of −100 mV, filtered at 10 kHz using the built-in low-pass Bessel filter, and digitized at 100 kHz with a Digidata 1550B interface and Clampex 10.7 software (Molecular Devices). Patch pipettes were pulled from thick-walled borosilicate glass (OD 1.5 mm; ID 0.87 mm; Hilgenberg, Germany) on a P-1000 puller (Sutter Instruments, Novato, CA, USA), giving resistances of 8–12 MΩ. Pipettes were Sylgard-coated (Dow Corning, Midland, MI, USA) and fire-polished to reduce noise.

The pipette/bath solution contained the following (in mM): 102.7 NaCl, 20 Na-gluconate, 2 KCl, 2 CaCl_2_, 1.2 MgCl_2_, 10 HEPES, 20 TEA-Cl, 14 glucose, and 15 sucrose (Carl Roth, Karlsruhe, Germany), adjusted to pH 7.4 with NaOH. The bath volume was reduced to 0.9–1 mL in 35 mm dishes to minimize noise. Only patches with a seal resistance >10 GΩ were analyzed. Unless stated, chemicals were from Merck (Darmstadt, Germany).

Single-channel clusters were identified visually. In addition to 10 kHz analog filtering, data were digitally filtered with an eight-pole Bessel filter in Clampfit to yield a 15:1 signal-to-noise ratio, and were down-sampled to maintain a 10:1 ratio between sampling frequency and effective cut-off (f_c_), calculated as (f_a_—analog filter; f_d_—digital filter cut-off): $\frac{1}{f_{c}}=\frac{1}{f_{a}}+\frac{1}{f_{d}}$.

Idealization was performed with SCAN (DCPROGS, http://www.onemol.org.uk/ (accessed on 2 February 2024)), using time-course fitting, with visual inspections of each trace. This method provided a 40–90 μs resolution for open and closed events, considerably superior to threshold detection and essential for GABA_A_Rs, which generate small-amplitude single-channel currents compared with other pLGICs. Idealization files (.scn) were used for the calculation of open/shut dwell time distributions in EKDIST and of kinetic modeling in HJCFIT.

### 4.4. Kinetic Modeling

The macroscopic current-based kinetic modeling was carried out in Python using the NumPy, Pandas, SciPy, and SCALCS (https://github.com/DCPROGS/SCALCS (accessed on 2 February 2024) and https://github.com/michal2am/bioscripts (accessed on 2 February 2024)) packages. As the initial scheme for WT GABA_A_Rs, we employed the model shown in Figure 4a, with rate constants adjusted to reproduce 500 ms responses to saturating [GABA], based on our previous studies with minor refinements (Table 3). From this reference model, simulations were performed in which individual rate constants were systematically varied to 0.05, 0.10, 0.15, 0.25, 0.50, 0.75, 1.25, 1.50, 1.75, 2.0, 2.25, 2.5, 2.75, 3.0, 3.25, 3.5, 3.75, 4.0, 5.0, 7.5, and 10.0 times their WT values, to examine the impact of the rates alterations on the receptor kinetics.

Single-channel–based kinetic modeling was performed using HJCFIT, a module of the DCPROGS package (http://www.onemol.org.uk (accessed on 2 February 2024) [27,28]), and Python scripts (https://github.com/michal2am/bioscripts, accessed on 2 February 2024). For the recordings from each cell and receptor type (WT and mutants), the rate constants of the model in Figure 4a (without the ligand binding steps) were estimated using the maximum likelihood method with correction for missed events. The model’s accuracy was further validated by comparing experimental dwell time distributions with those predicted from the fitted parameters.

### 4.5. Sequence and Structure Analysis

Structural analyses and visualizations were performed using the α_1_β_2_γ_2_ GABA_A_R receptor structure bound to GABA (PDB ID: 6X3Z, Kim et al., 2020 [11]) within the ChimeraX environment [29] and VMD [30]. The α_1_β_2_γ_2_ GABA_A_R β_2_ E153A-mutant structure for molecular interaction analysis in FoldX [24] was prepared in CHARMM-GUI [31] using its PDB Manipulator [32,33,34] and Membrane Builder [35,36]. The protein sequences of the GABA receptor’s subunits were retrieved from the UniProt database [37]. Sequence alignments were initially generated with T-Coffee [38,39] and subsequently refined manually in JalView [40], which was also used to prepare the alignment figures.

### 4.6. Statistical Data Analysis and Visualization

Data were initially organized and processed in Microsoft Excel, while more advanced analyses and visualizations were carried out using custom Python scripts. Outliers were identified based on normalized z-scores. The normality of data distributions was examined with the Shapiro–Wilk test, and, where appropriate, the equality of variances was verified using Levene’s test. Depending on whether the data met the assumptions of normality, the statistical significance of mutation effects on the channel kinetics or event dwell time distributions was assessed using either one-way ANOVA or the nonparametric Kruskal–Wallis H-test. Post hoc pairwise comparisons were performed with Tukey’s test or Dunn’s test with Holm–Šidák correction. All statistical analyses were implemented in Python utilizing the Pandas, NumPy, and SciPy libraries, and visualizations were generated using the Seaborn package.

## Figures and Tables

**Figure 1 ijms-27-00047-f001:**
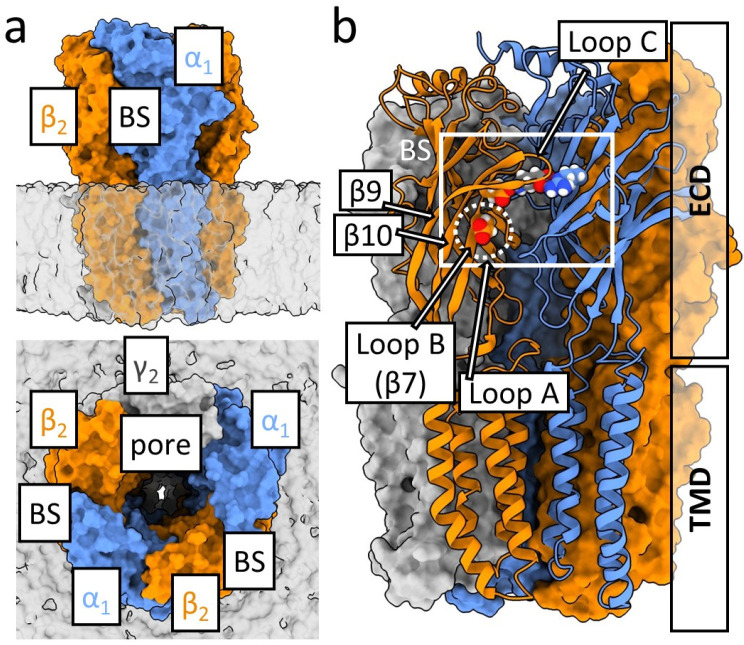
Structure of the GABA_A_R. (**a**) Surface visualization of the GABA_A_R (α_1_ subunit in blue, β_2_ in orange, and γ_2_ in gray) in the lipid membrane (transparent). GABA binding sites (BS) are located at β_2_/α_1_ subunit interfaces; ion pore is formed by all five subunits. (**b**) Side view if the GABA_A_R with marked ECD and TMD domains. GABA binding site region is shown in white rectangle. Agonist molecule (capped below the loop C) is stabilized electrostatically by β_2_E155 and α_1_R67 shown in spherical visualizations. β_2_E153 is located below the agonist marked in dotted circle.

**Figure 2 ijms-27-00047-f002:**
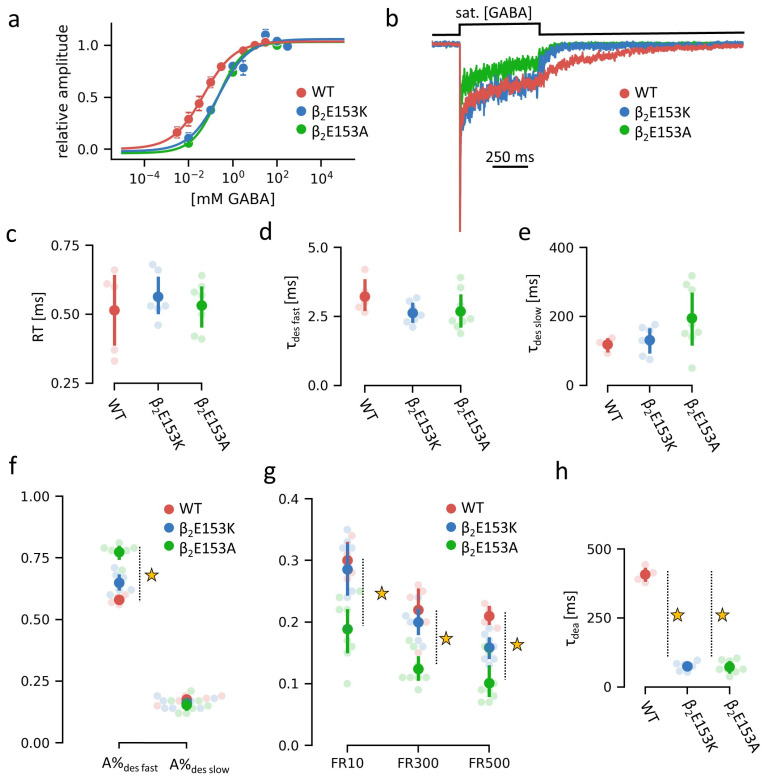
Macroscopic recordings of the WT and β_2_E153 mutants of GABA_A_R-mediated currents evoked by rapid applications of GABA (numeric values of presented data in Table 1). (**a**) Dose–response curve. (**b**) Exemplary recorded traces showing the receptor response to 500 ms pulse of saturating [GABA]. (**c**–**h**) Plots showing the kinetic parameters of the current responses for the WT and β_2_E153 mutants: the rise time, desensitization time constants (fast and slow), percentages of fast and slow desensitization components, fractions of remaining current after 10, 300 and 500 ms of start of agonist application, and deactivation time constant, respectively. The average values of parameters are presented with dots, with marked 0.95 confidence intervals. Values differing from WT at *p*-value < 0.05 are starred.

**Figure 3 ijms-27-00047-f003:**
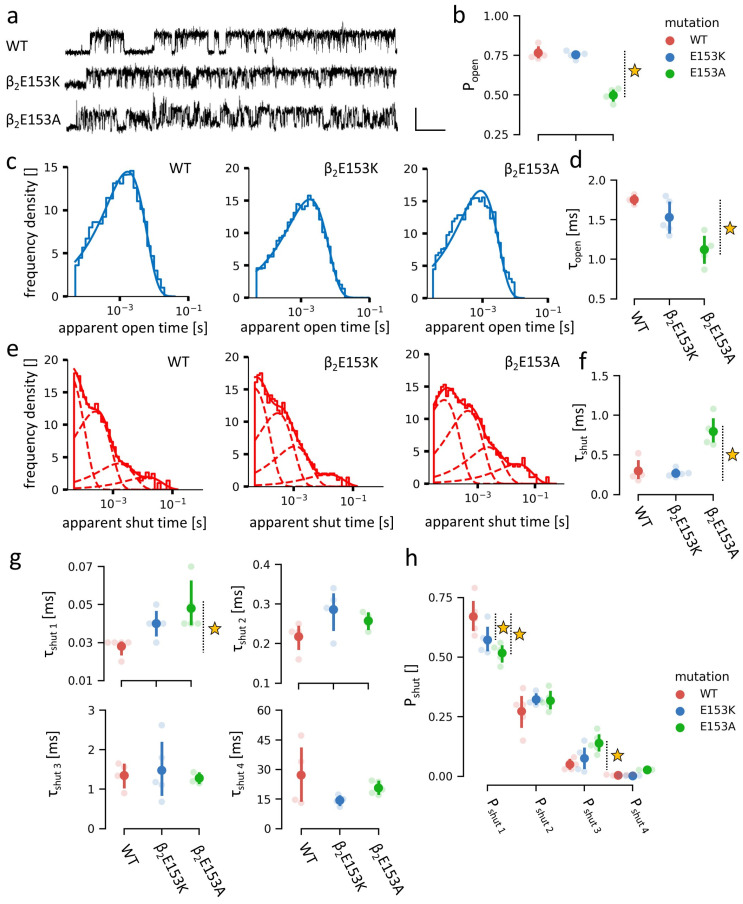
Single channel recordings of the WT and β_2_E153 mutants of GABA_A_Rs (numeric values of presented data in Table 2). (**a**) Exemplary recorded traces showing the receptor’s activity at saturating [GABA] in stationary conditions. (**b**) Plot showing single-channel opening probability measured in clusters. (**c**) Exemplary (recoding from a single typical cell) open dwell time histograms of open state for WT and β_2_E153-mutant GABA_A_Rs. (**d**) Plot showing the time constant of the open distributions (indicative for mean dwell time of open state). (**e**) Exemplary (recoding from single typical cell) dwell time histograms of shut states of WT and β_2_E153 mutants GABA_A_Rs. Dashed lines show respective components. (**f**) Plot showing the mean dwell time of shut states. (**g**,**h**) Plots showing detailed values of parameters (time constants and ratios of components) of the shut state dwell time distribution (weighted average in (**f**)). The average values of parameters are presented with dots, with marked 0.95 confidence intervals. Values differing from WT at *p*-value < 0.05 are starred.

**Figure 4 ijms-27-00047-f004:**
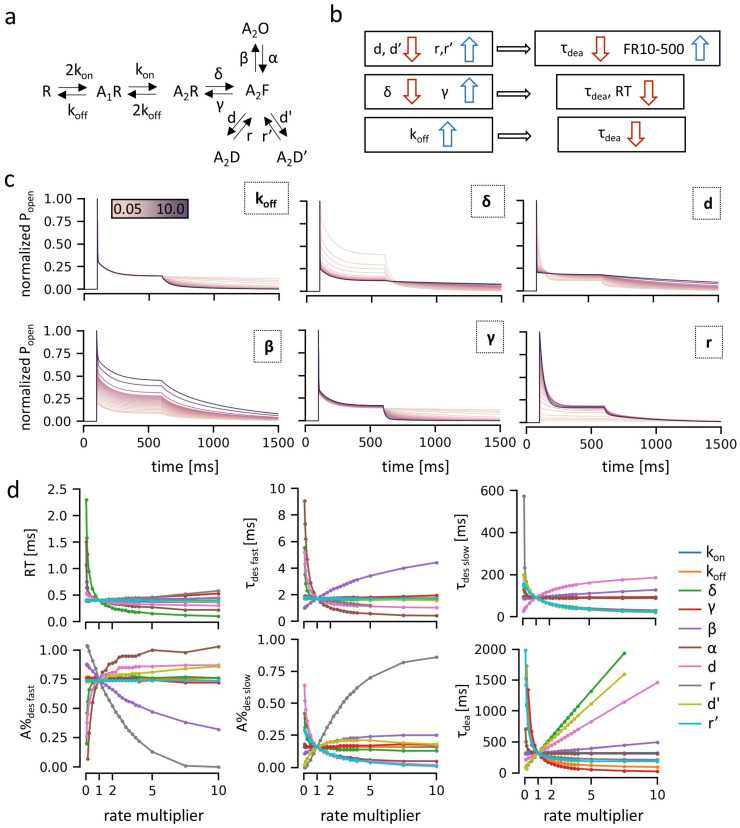
Kinetic modeling of the macroscopic responses of the GABA_A_Rs to saturating [GABA]. (**a**) Scheme of the kinetic model used for simulations: R—resting state, A_1_R—single agonist bound state, A_2_R—double agonist bound state, A_2_F—flipped state, A_2_D and A_2_D’—short- and long-lived desensitization states, and A_2_O—open state. Symbols at arrows indicate transition rates (numeric values in Table 3). (**b**) Most parsimonious scenarios allowing the reproduction of the deactivation kinetics acceleration; arrows indicate changes in the kinetic model rates and resulting changes in parameters describing kinetics of current responses. (**c**) Simulated traces of the GABA_A_R macroscopic response to saturating [GABA] 500 ms pulse. In each plot, simulations for the given kinetic parameter alterations (scaling by multiplier values in the range from 0.05 to 10.0) are presented. The amplitudes of simulated traces are normalized to 1.0. (**d**) Plots showing the dependence of the kinetic parameters of the macroscopic responses on the model rates alterations.

**Figure 5 ijms-27-00047-f005:**
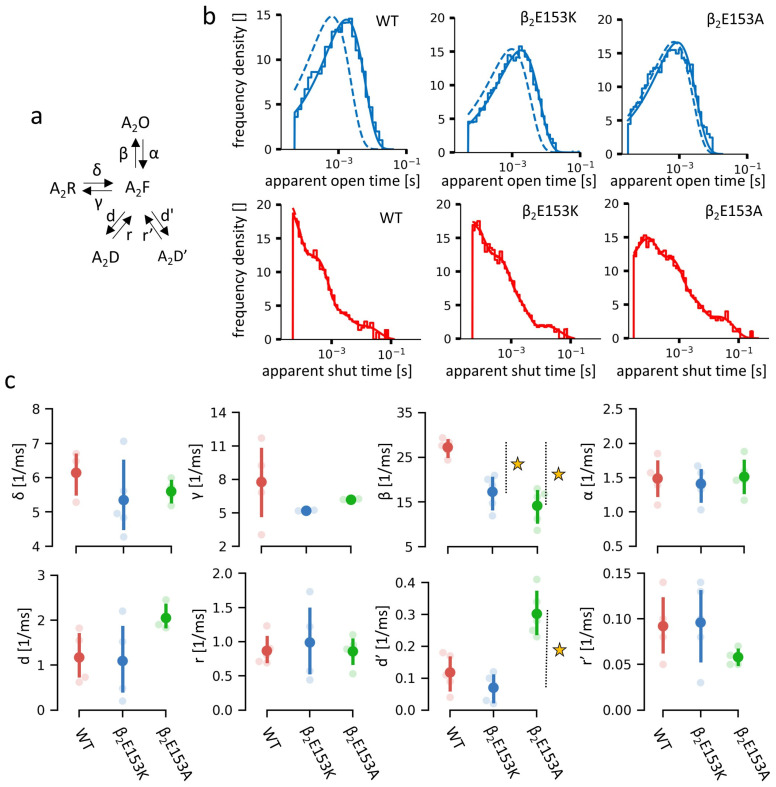
Kinetic modeling of single channel activity of the GABA_A_Rs at saturating [GABA] in the equilibrium conditions. (**a**) Scheme of the kinetic model used for simulations. States’ names are the same as in Figure 4a. (**b**) Exemplary (recoding from a single typical cell) dwell time histograms of open and shut states of WT and β_2_E153-mutant GABA_A_Rs: histograms represent experimental data; solid and dashed fit lines represent the distribution based on fitted model parameters at experimental and at ideal (equal to zero, with missed event correction) recording resolutions. (**c**) Plots showing the values of the fitted model parameters for WT and respective β_2_E153 GABA_A_R mutants. The average values of parameters are presented with dots, with marked 0.95 confidence intervals. Values differing from WT at *p*-value < 0.05 are starred (numeric values of presented data in Table 4).

**Figure 6 ijms-27-00047-f006:**
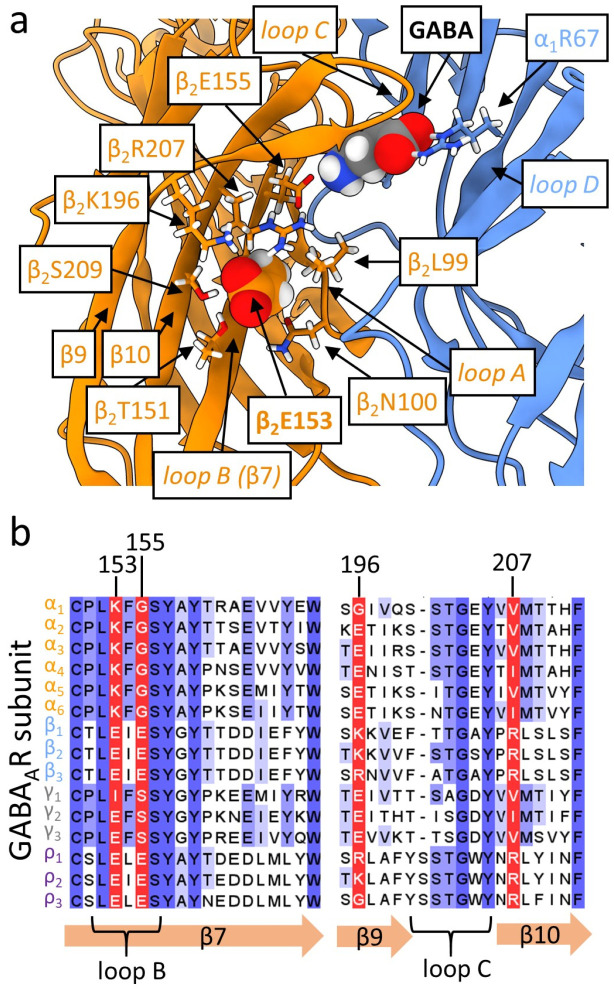
Structure and sequence of the β_2_E153 region in GABA_A_R. (**a**) Detailed view of the β_2_E153 (showed in spherical representation) area. GABA molecule is located between β_2_E155 and α_1_R67. β_2_E153 neighboring residues are shown in stick representation. (**b**) Alignment of GABA_A_R α, β, γ, and ρ subunit sequences. Discussed residues are marked red; others are colored in blue according to conservation level (darkest shade—most conserved).

## Data Availability

The data presented in this study are openly available in Zenodo at 10.5281/zenodo.17491460.

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
