# Peer review of "β_2_E153 Residue at Loop B of GABA_A_R Is Involved in Agonist Stabilization and Gating Properties"

_ijms, 2025, doi:10.3390/ijms27010047_

Round 1
Reviewer 1 Report
Comments and Suggestions for Authors
In this manuscript, Michalowski et al. presented eletcrophysiology and modelling studies of GABAA receptor beta2 subunit E153 residue. The researrch is well designed, and the results are well presented. However, I find this paper contains only little new information. The effects and importance of E153 residue and mutations have been very well documented (see Venkatachalan, Srinivasan P., and Cynthia Czajkowski. "A conserved salt bridge critical for GABAA receptor function and loop C dynamics." Proceedings of the National Academy of Sciences 105, no. 36 (2008): 13604-13609. and Michałowski, Michał A., Katarzyna Terejko, Michalina Gos, Ilona Iżykowska, Marta M. Czyżewska, Karol Kłopotowski, Przemysław T. Kaczor et al. "Φ value analysis underscores strong functional and structural compactness of the GABAA receptor." Proceedings of the National Academy of Sciences 122, no. 38 (2025): e2512278122. , the second being the author's own paper.)
I recommend the authors include more data on the kinetics modelling of this paper so that the paper is different and provide more new insight from published material.
Author Response
Comments 1: In this manuscript, Michalowski et al. presented eletcrophysiology and modelling studies of GABAA receptor beta2 subunit E153 residue. The researrch is well designed, and the results are well presented. However, I find this paper contains only little new information. The effects and importance of E153 residue and mutations have been very well documented (see Venkatachalan, Srinivasan P., and Cynthia Czajkowski. "A conserved salt bridge critical for GABAA receptor function and loop C dynamics." Proceedings of the National Academy of Sciences 105, no. 36 (2008): 13604-13609. and Michałowski, Michał A., Katarzyna Terejko, Michalina Gos, Ilona Iżykowska, Marta M. Czyżewska, Karol Kłopotowski, Przemysław T. Kaczor et al. "Φ value analysis underscores strong functional and structural compactness of the GABAA receptor." Proceedings of the National Academy of Sciences 122, no. 38 (2025): e2512278122. , the second being the author's own paper.) I recommend the authors include more data on the kinetics modelling of this paper so that the paper is different and provide more new insight from published material.
Response 1: We thank the reviewer for this comment and for highlighting previous investigations regarding the role of the E153 residue. We fully agree that E153 has been recognized as an important determinant of GABAAR function; however, the present study provides several layers of new information that go beyond what has been previously reported.
First of all, what was essentially lacking thus far was a thorough description of the impact of the E153 residue on the receptor binding and gating properties. In the important paper by Venkatachalan et al. (2008), the authors addressed local interactions of this residue and describe the effect of its mutation of the receptor activation without specifying to what extent this residue is involved in agonist binding and gating of the bound receptor. Indeed, they state in the Results section that “The EC 50 values used in our analysis are a composite of microscopic agonist binding and channel gating constants. This complicates the analysis and our ascribing whether the interaction influences agonist binding and/or gating.”. This limitation resulted from the experimental model they used (oocytes and two electrodes voltage-clamp) which allows for time resolution in the range of seconds, precluding thus detailed kinetic and pharmacologic analysis. To fill this gap we have integrated high–temporal-resolution (submillisecond, 40-90 ms) single-channel recordings with detailed kinetic modeling applied for these data. This approach allowed us to resolve gating transitions with precision adequate to the time scale of physiological phenomena such as synaptic currents. The macroscopic recordings included in our manuscript were also performed with high temporal resolution, providing additional complementary information enabling, in particular, a reliable quantification of fast components of activation, macroscopic desensitization and deactivation. In addition, our receptor paper (Michałowski et al. 2025, PNAS) does not provide any detailed description of the role of the E153 residue in the receptor gating – in this report only some specific gating parameters (for WT and E153 mutants) were applied to calculate the F value. We are thus convinced that our study extends previous description of the role of E153 residue by providing a thorough characterization of its impact on gating properties of GABAA receptor.
In response to the reviewer’s comment, we extended the structural analysis to directly test the proposed mechanism. To provide a support for the hypothesis outlined in Discussion (lines 347-351), we prepared the model of the β2E153A mutant using the WT structure and estimated the changes in the interaction energy between fragments of the β-strands contacting residues β2E153 and β2K196 using the FoldX suite. In each β2 subunit, the mutation to alanine made this interaction roughly twofold less energetically favorable, consistent with a destabilization of the local β-sheet network. We would like to emphasize that the structural interpretation presented in this study is directly anchored in the most up-to-date structural data available for GABAARs By combining single-channel analysis, kinetic modeling, and current high-resolution structures, we provide mechanistic insights that were not accessible in earlier work, including that of Venkatachalan et al. (2008) and our own previous Φ-value analysis (Michałowski et al., 2025). Importantly, this integrated approach reveals how mutations at position E153 perturb specific transitions within the receptor’s activation landscape, thereby supporting the conclusions drawn from the functional experiments.
Changes in the manuscript:
Discussion section (starting at line 351):
To verify this hypothesis we prepared the model of the β2E153A mutant using the WT structure [11] and estimated the changes in the interaction energy between fragments of the β-strands with residue β2E153 and β2K196 using FoldX [24] suite. As expected, in case of each β2 subunit after the mutation to alanine this interaction was roughly twice less energetically favorable.
Methods section (starting at line 500):
Structural analyses and visualizations were performed using the α1β2γ2 GABAAR receptor structure bound to GABA (PDB ID: 6X3Z, Kim et al., 2020) within the ChimeraX environment [28] and VMD [29]. α1β2γ2 GABAAR β2 E153A mutant structure for molecular interaction analysis in FoldX [30] was prepared in CHARMM-GUI [31] using its PDB Manipulator [32–34] and Membrane Builder [35,36].
Reviewer 2 Report
Comments and Suggestions for Authors
This manuscript investigated how β2E153 residue at Loop B of GABAAR is involved in agonist stabilization and gating properties through patch-clamp electrophysiological recordings combined with mutations. The authors conclude that β2E153 is a key element in the long-range allosteric network to link the binding site to the channel gate in GABAARs. In general, the results are potentially interesting.
Specific comments:
- The figure 1 is listed in the introduction and background part, which makes one wonder whether this figure is from the authors or it is a copy-and-paste figure. If the figure is not made by the authors, please clearly state it in the figure legend and cite it appropriately.
- It still remains unclear whether the mutation of E153 would affect one of most fundamental properties of the receptor, the ion-permeable conductance.
- In the ionic recordings such as Figure 3A, it is not clear what the recording voltage is. Please state it clearly so that the data quality can be accurately assessed.
- In the table 1, the EC50 value varies among these three variants (including the WT). However, in figure 3B, the Popen does not look different between red and blue, which makes one wonder whether the concentration of GABA applied was exactly same for all three variants. If not, would the comparison still be meaningful? If yes, why the Popen look so similar?
- Please check the capital letter of A/B/C etc in the figure legend and make sure that they are consistent with the figures.
Author Response
Comments 1: The figure 1 is listed in the introduction and background part, which makes one wonder whether this figure is from the authors or it is a copy-and-paste figure. If the figure is not made by the authors, please clearly state it in the figure legend and cite it appropriately.
Response 1: We confirm that Figure 1 was created entirely by the authors for the purposes of this study.
Comments 2: It still remains unclear whether the mutation of E153 would affect one of most fundamental properties of the receptor, the ion-permeable conductance.
Response 2: Based on our single-channel recordings, we did not observe any evidence that mutations at β2E153 alter the ion-permeation conductance of the receptor. All recordings were idealized using the SCAN software, in which each channel opening and closing event, along with its current amplitude, requires manual verification by the user. Throughout this process, we did not detect any openings with systematically increased or decreased amplitudes that would indicate changes in pore conductance. We have added a statement clarifying this point in the revised manuscript (line 160).
Changes in the manuscript (starting at line 160):
In both mutants, the amplitudes of the single-channel currents were comparable to those of the WT receptor, indicating that the mutations did not alter the channel conductance. In addition, in the case of mutations no conductance sublevels or extra large current amplitudes were observed.
Comments 3: In the ionic recordings such as Figure 3A, it is not clear what the recording voltage is. Please state it clearly so that the data quality can be accurately assessed.
Response 3: We thank the reviewer for pointing this out. The macroscopic recordings were performed at a holding membrane voltage of –40 mV, as stated in the Methods section (line 422). Single-channel recordings were conducted at a pipette potential of –100 mV; this information was previously missing and has now been added to the revised Methods section (line 460) for clarity.
Changes in the manuscript (starting at line 460):
Currents were amplified with an Axopatch 200B amplifier (Molecular Devices, Sunnyvale, CA, USA) at a holding pipette potential of −100 mV, filtered at 10 kHz using the built-in low-pass Bessel filter, and digitized at 100 kHz with a Digidata 1550B interface and Clampex 10.7 software (Molecular Devices).
Comments 4: In the table 1, the EC50 value varies among these three variants (including the WT). However, in figure 3B, the Popen does not look different between red and blue, which makes one wonder whether the concentration of GABA applied was exactly same for all three variants. If not, would the comparison still be meaningful? If yes, why the Popen look so similar?
Response 4: We appreciate the reviewer’s thoughtful question. The EC₅₀ values reported in Table 1 and the dose–response curves shown in Figure 1 were obtained from whole-cell macroscopic recordings across a range of GABA concentrations. These dose–response relationships were then used to determine the saturating agonist concentration, which was 10 mM for the WT and for both mutants. All subsequent macroscopic excised-patch and single-channel experiments were therefore performed at this saturating concentration, as noted in the manuscript (lines 112 and 144). The Popen values presented in Figure 3B were calculated from selected clusters of channel activity (line 155). Within these clusters, receptors are assumed to be fully occupied by agonist due to the saturating GABA concentration in the equilibrium conditions. Under such conditions, Popen is no longer reflecting the changes in the EC₅₀ or in agonist-binding kinetics; instead, it reflects differences in the receptor activation including transitions such as flipping, gating, or desensitization. Thus, EC₅₀ values and cluster-based open estimates capture distinct aspects of receptor function, which explains why the variants may differ in EC₅₀ yet display similar Popen values under saturating conditions.
Changes in the manuscript (starting at line 157):
In clusters, at the saturating GABA concentration receptors are assumed to be in the fully bound state, thus the changes in the open probability indicate alterations in the receptor transitions other than binding and do not directly reflect the alterations in EC50 value
Comments 5: Please check the capital letter of A/B/C etc in the figure legend and make sure that they are consistent with the figures.
Response 5: We thank the reviewer for noting this inconsistency. The panel labels have now been revised to use lowercase letters, in accordance with the journal’s formatting requirements, and are fully consistent with the corresponding figure legends.
Round 2
Reviewer 1 Report
Comments and Suggestions for Authors
The authors have addressed my concerns.
Reviewer 2 Report
Comments and Suggestions for Authors
The authors have addressed my original concerns and questions satisfactorily and made revisions accordingly.